# Quantitative analysis of medical students' and physicians' knowledge of degenerative cervical myelopathy

Mueez Waqar  ,[1] Jane Wilcock,[2] Jayne Garner,[2] Benjamin Davies,[3] Mark Kotter[3,4]

[1]Academic Neurosurgery, University of Manchester, Manchester, UK
[2]Department of Undergraduate Medical Education, University of Liverpool, Liverpool, UK
[3]Department of Clinical Neurosurgery, University of Cambridge, Cambridge, UK
[4]Department of Clinical Neurosciences, Ann McLaren Laboratory of Regenerative Medicine, Cambridge, UK

**Correspondence to**
Dr Mark Kotter;
mrk25@cam.ac.uk

## ABSTRACT

**Objectives** We have previously identified a delay in general practitioner (GP) referrals for patients with degenerative cervical myelopathy (DCM). The aim of this study was to evaluate whether an education gap existed for DCM along the GP training pathway by quantitatively assessing training in, and knowledge of, this condition.

**Design** Gap analysis: comparison of DCM to other conditions. Comparators selected on the basis of similar presentation/epidemiology (multiple sclerosis), an important spinal emergency (cauda equina syndrome) and a common disease (diabetes mellitus).

**Subjects** Medical students, foundation doctors and GP trainees.

**Primary and secondary outcome measures** (1) Assessment of training: quantitative comparison of references to DCM in curricula (undergraduate/postgraduate) and commonly used textbooks (Oxford Handbook Series), to other conditions using modal ranks. (2) Assessment of knowledge: using standardised questions placed in an online question-bank (Passmedicine). Results were presented relative to the question-bank mean (+/−).

**Results** DCM had the lowest modal rank of references to the condition in curricula analysis and second lowest modal rank in textbook analysis. In knowledge analysis questions were attempted 127 457 times. Performance for DCM questions in themes of presentation (+6.1%), workup (+0.1%) and management (+1.8%) were all greater than the question-bank mean and within one SD. For students and junior trainees, there was a serial decrease in performance from presentation and workup (−0.7% to +10.4% relative to question-bank mean) and management (−0.6% to −3.9% relative to question-bank mean).

**Conclusions** Although infrequently cited in curricula and learning resources, knowledge relating to DCM was above average. However, knowledge relating to its management was relatively poor.

## INTRODUCTION

Degenerative cervical myelopathy (DCM) is a common and insidious condition that can lead to severe disability.[1] It arises when degenerative changes of the spine compress the spinal cord, causing a progressive spinal cord injury. Currently, treatment is limited to surgical decompression, which is able to stop further injury but due to the limited

### Strengths and limitations of this study

► Search terms relating to degenerative cervical myelopathy (DCM) were queried from three UK specific medical school curricula and relevant postgraduate curricula.
► A large number of responses were obtained by placing questions in an online question-bank, relating to DCM.
► A limited number of learning resources were searched to assess references to DCM.

capacity of the spinal cord to repair, recovery is limited. The timing of surgery is therefore crucial to recovery ('time is spine'), and a recent meta-analysis has demonstrated treatment within 6 months offers a greater chance of making a full recovery.[2] Unfortunately, few patients are diagnosed promptly, with the majority waiting more than 2 years for a diagnosis.[3] Consequently, most patients retain life-long disabilities, contributing to quality of life scores lower than cancer, heart and lung diseases.[4 5]

The diagnostic pathway for DCM almost exclusively starts with assessment and triage by a community physician, termed in the UK a general practitioner (GP).[3] If appropriate, the patient is then referred on for further investigation and management. Our analysis of this pathway has identified time to initial referral by GP (6.4±7.7 months) as representing 51% of diagnostic delay.[6]

This period of the diagnostic pathway is difficult to examine in detail, and while delayed patient presentation is likely to contribute, delayed detection measured by multiple consultations and patient perspective, is certainly a relevant component.[3 7] The cause for delayed detection is also likely to be multi-faceted, including subtle, non-specific symptoms and incomplete clinical examination.[1] However, it is conceivable that a lack of understanding of the workup and management of DCM may also contribute.

Indeed, DCM would be included within the spectrum of 'neurophobia'—an aversion to the neurosciences due to perceived difficulty, which has been demonstrated in GPs and GP trainees.[8 9]

Our objective therefore was to evaluate whether an education gap exists for DCM among GPs, by quantitatively assessing their training in, and knowledge of, DCM.

## METHODS
### Patient and public involvement
DCM patients were surveyed online and this confirmed that the diagnosis of DCM is frequently delayed.[10] The question of whether this is due to lack of knowledge among health professionals and whether this was the consequence of a gap in medical education was formulated with the input of DCM sufferers at the First Cambridge Myelopathy day.

### Education gap analysis
A gap analysis is a process to identify gaps in existing systems such as education curricula. It involves an assessment of existing knowledge or standards against predetermined standards that define core competency requirements.[11 12] This differs, for example, from simple cross-sectional knowledge assessments whose primary outcome is usually assessment of knowledge beyond core requirements.

The study objective was therefore approached in two separate gap analyses (table 1).
1. *Assessment of training*: quantification of DCM in curricula and commonly used learning resources, including assessment of relative importance through comparison to other conditions.
2. *Assessment of knowledge*: formal assessment of trainees' knowledge using questions placed in an online question-bank.

### GP training pathway
UK medical graduates complete the UK Foundation Programme to enter higher specialty training as formal GP trainees. There are several assessments along this route including: medical school final examinations to gain registration to practice in the UK, or the equivalent Professional and Linguistic Assessments Board (PLAB) test for international medical graduates; the Specialty Recruitment Assessment (SRA)—a written assessment taken towards the start of the second foundation year prior to higher specialty applications; and the Membership of The Royal College of General Practitioners (MRCGP), an exit examination for GP trainees. While alternative entry routes to general practice exist, this pathway represents the most common training route for GPs today.

### Definition of DCM and comparators
DCM was chosen as an inclusive term for a variety of diseases resulting in compressive myelopathy.[1] Comparator diseases were also selected to compare findings to:
1. *Direct comparator to myelopathy*: a disease that is a differential diagnosis for DCM with equivalent or greater incidence.
2. *Degenerative spine comparator*: an alternative degenerative spine disease that is widely taught.
3. *Generic non-neuroscience comparator*: a common disease that all clinicians would have some knowledge about and interaction with.

The a priori hypothesis was, if an education gap existed within DCM, we would expect metrics of training and knowledge of DCM to rank inferior to controls. Controls were identified through a consensus author meeting and finalised after confirmation of epidemiological profiles through literature review.

### Assessment of training: learning resource analysis
We selected learning resources that were in common usage by trainees, available in electronic format (to be amenable for searching) and stratifiable by training stage. Specifically, training curricula, the Oxford Handbook series[13] and online question-banks were selected. Alongside the UK Foundation Programme and MRCGP, an example from each of the three main, UK medical school teaching models[14] were included: problem-based learning (The University of Manchester, UK); traditional

**Table 1** Summary of gap analysis methods

| | Medical school | Foundation | GP training | Metric |
|---|---|---|---|---|
| Curricula | UoM UoC Imperial | Foundation programme curriculum | MRCGP curriculum | References to search terms |
| Text book | OHCM | OHFP | OHGP | References to search terms |
| Online question-bank | PLAB Medical finals | SRA | AKT | Performance in questions |

This table shows the methods used in this study. Curricula and textbooks were screened by training stage to assess references to key search terms. An online question-bank was used in knowledge assessment.

AKT, applied knowledge test; GP, general practitioner; MRCGP, Membership of The Royal College of General Practitioners; OHCM, *Oxford Handbook of Clinical Medicine*; OHFP, *Oxford Handbook for the Foundation Programme*; OHGP, *Oxford Handbook of General Practice*; PBL, problem based learning; PLAB, Professional and Linguistic Assessments Board; SRA, Specialty Recruitment Assessment; UoC, University of Cambridge; UoM, University of Manchester.

lecture based (The University of Cambridge, UK); an integrated system including aspects of both (Imperial College London, UK). The choice of medical school curricula was pragmatically selected, based on access. The handbooks most pertinent to the GP training pathway were included in our analysis: the *Oxford Handbook of Clinical Medicine*, the *Oxford Handbook for the Foundation Programme* and the *Oxford Handbook of General Practice.*

Search terms for analysing learning resources were created for DCM and each comparator using relevant terms from the search syntax of Cochrane Reviews, or if absent, recent systematic reviews. Curricula were searched for the number of references per disease, text books for the number of words per disease section and question-banks for the number of questions.

### Assessment of knowledge

The author panel includes GPs with an interest in education (JW), university appointed educationalists (JG) and neurosurgeons (MW, BD, MK). The authors devised a set of questions to be included in an online question-bank, composed both multiple choice questions (MCQs) and extended matching questions (EMQs), designed to cover the different components of medical assessment; presentation, investigation and management. The final set consisted of 19 questions (13 MCQs and six EMQs). The questions together with answers are included in online supplementary 1.

Online question-banks including subsections devoted to each stage of the GP training pathway were contacted with details of the study. Only one question-bank—Passmedicine,[15] responded and was therefore selected for this arm of the study.

### Analysis

Statistical analysis was performed using SPSS V.22. For assessment of training, frequencies of search terms were described and the mode and modal ranks determined. For assessment of knowledge, histograms were constructed with either a condition or question-bank along the x-axis and user performance relative to question-bank mean along the y-axis. The user performance relative to question-bank mean corresponded to the raw difference between the mean value for each theme and the mean value for a particular question-bank.

## RESULTS
### Selection of comparators and resources

Multiple sclerosis (MS), cauda equina syndrome (CES) and diabetes mellitus were selected as the three disease comparators and list of search terms compiled accordingly (table 2).[16–19] MS is a common differential for DCM,[20] with overlapping signs and symptoms. While DCM is likely to be more prevalent in reality,[1] currently their characterised epidemiology is comparable.[21 22] CES is uncommon, but its missed or delayed diagnosis carries significant consequences. Diabetes is relevant to all medical fields. The question-bank also provided data on neurology as a theme, encompassing all questions relating to the central and peripheral nervous system.

### Assessment of training
#### Curricula analysis

DCM and CES had the lowest modal rank in curricula search analysis (table 3). This was true for early stage undergraduate curricula and late stage curricula for the foundation programme and MRCGP. MS was mentioned less frequently in early versus late stage curricula. The opposite trend was observed for diabetes mellitus.

#### Textbook analysis

Overall, DCM had the second lowest modal rank but above that of CES (table 4). The relative word count attributed to neurological conditions such as DCM, MS and CES decreased with advancing stage of textbook. The opposite trend was observed for diabetes mellitus, for whom the total word count increased.

### Question bank analysis
#### Assessment of knowledge

Passmedicine was approached, on the basis it did not have any DCM questions previously which could influence analysis. Questions were introduced between the period June 2017 and October 2017. Section editors for each question-bank ultimately selected which of the 19 questions were relevant, such that finals/PLAB contained 14, SRA—19 and MRCGP—19.

Overall, questions were attempted 127 457 times; finals/PLAB—36 706; SRA—47 530; MRCGP—43 221 (online supplementary table 1). Data on the number of unique attempts or first-time viewers was not extractable.

There were differences in user performance in the three DCM question themes—presentation, workup and management, though average scores were all within one SD of the mean (figure 1). Performance sequentially decreased across these themes for the finals/PLAB group (presentation +4.7%, workup +4.2%, management −0.6%; relative to question-bank mean), SRA group (presentation +10.4%, workup −0.7%, management −3.9%; relative to question-bank mean) and when all groups were considered together (presentation +6.1%, workup +0.1%, management +1.8%; relative to question-bank mean). In the MRCGP group however, performance clustered around the mean for the question-bank with little variability (presentation −0.1%, workup +1.0%, management +1.8%; relative to question-bank mean).

User performance by disease showed large variation between question-banks (figure 2). The mean user performance decreased sequentially for DCM with advancing question-bank (finals/PLAB +4.2%, SRA +2.7%, MRCGP +0.9%; relative to question-bank mean). However, the mean user performance for DCM was always greater than the question-bank mean (+2.6% relative to question-bank mean).

**Table 2** Selected diseases for textbook and curricula searches

| | Degenerative cervical myelopathy | Multiple sclerosis | Cauda equina syndrome | Diabetes mellitus |
|---|---|---|---|---|
| Reasoning | Disease of interest | A disease with a similar incidence and morbidity to degenerative cervical myelopathy. Selected as a clinico-epidemiological control[22] | Widely taught spinal emergency, selected as a spinal control | Non-spinal control |
| Search terms for curricula searches | ▶ Cervical myelopathy.<br>▶ Cervical myeloradiculopathy.<br>▶ Cervical stenosis.<br>▶ Cervical compression.<br>▶ Cervical herniation.<br>▶ Cervical degeneration.<br>▶ Ossification of posterior longitudinal ligament.<br>▶ Spinal osteophytosis.<br>▶ Spinal cord compression.<br>▶ Spondylosis. | ▶ Multiple sclerosis.<br>▶ Demyelinating disease.<br>▶ Demyelination. | ▶ Cauda equina.<br>▶ Saddle anaesthesia. | ▶ Diabetes mellitus.<br>▶ Insulin dependent diabetes mellitus.<br>▶ Type 1 diabetes mellitus.<br>▶ Non-insulin dependent diabetes mellitus.<br>▶ Type 2 diabetes mellitus.<br>▶ Maturity onset diabetes of young.<br>▶ Gestational diabetes mellitus.<br>▶ Late onset diabetes.<br>▶ Maturity onset diabetes.<br>▶ Insulin resistance.<br>▶ Diabetic ketoacidosis.<br>▶ Hyperosmolar hyperglycaemic state.<br>▶ Hyperosmolar non-ketotic coma. |

The reasoning for each of the comparators and search terms employed for the curricula searches are explained in this table.

## DISCUSSION

In this study, we found that DCM is relatively infrequently cited in curricula and commonly used textbooks, even compared with diseases such as MS, with similar epidemiology. Despite this however, user performance in some of the DCM questions remained consistently above question-bank averages. There was a sequential decrease in user performance across the themes of DCM presentation, workup and management for early years' trainees, whereas for senior trainees, performance did not vary by theme of question. For other neuroscience themes such as MS and CES, the user performance was below average. This was in contrast to user performance in questions relating to diabetes mellitus, which was consistently above question-bank mean.

We observed a below average performance in questions grouped under the neurology theme, which included all questions relating to the central and peripheral nervous

**Table 3** Curricula analysis

| | Number of references to term | | | |
|---|---|---|---|---|
| Curriculum | Degenerative cervical myelopathy | Multiple sclerosis | Cauda equina syndrome | Diabetes mellitus |
| Undergraduate—PBL | 0 | 3 | 1 | 43 |
| Undergraduate—traditional | 1 | 13 | 1 | 66 |
| Undergraduate—integrated | 3 | 15 | 0 | 30 |
| Foundation programme | 0 | 0 | 0 | 0 |
| MRCGP | 0 | 4 | 2 | 113 |
| Total | 4 | 32 | 4 | 252 |
| Rank | 3 | 2 | 3 | 1 |

Electronic copies of curricula were queried with relevant search terms (as shown in table 2).
MRCGP, Membership of The Royal College of General Practitioners.

**Table 4** Learning resource analysis

| Resource | Word count devoted to | | | |
| --- | --- | --- | --- | --- |
| | Degenerative cervical myelopathy | Multiple sclerosis | Cauda equina syndrome | Diabetes mellitus |
| OHCM | 870 | 1104 | 567 | 5165 |
| OHFP | 0 | 112 | 58 | 3599 |
| OHGP | 252 | 679 | 120 | 5736 |
| Cumulative | 1122 | 1895 | 745 | 14 500 |
| Modal rank | 3 | 2 | 4 | 1 |

The number of words devoted to degenerativecervical myelopathy and other diseases were determined. Importantly, only the words contained within the main section for the particular disease were considered.
OHCM, *Oxford Handbook of Clinical Medicine*; OHFP, *Oxford Handbook for the Foundation Programme*; OHGP, *Oxford Handbook of General Practice*.

system. Reduced knowledge pertaining to neurosciences has previously been linked to a term called neurophobia, though this is by no means a universally accepted concept.[8 9 23] In one questionnaire study for example, GPs rated neurology as the most difficulty medical specialty and the one for which they had the least confidence compared with cardiology, endocrinology, gastroenterology, geriatrics, respiratory medicine and rheumatology.[9] However, the question-bank data did not allow distinction between basic science and clinical questions, for whom the performance may be different, as evidenced by the above average performance in our clinically orientated DCM questions.

Second, our study supports the observation of poorer performance in management themes by clinicians at an earlier training stage. Prior studies on medical students

that have evaluated knowledge formally have shown similar results in specialties such as ophthalmology.[24] This also appears to translate into patient encounters during clinical assessments in neurology.[25 26] In one study evaluating student performance in neurology outpatients, students were more likely to make errors regarding diagnostic tests or planning treatment than lesion localisation.[25] Another study evaluating student performance in neurology themed objective structured clinical examinations (OSCEs) also reported similar results, finding poorer student performance in supportive management of neurological conditions versus diagnosis.[26]

Our original hypothesis attempting to correlate delays in diagnosis for DCM with a deficiency in training and knowledge, was not conclusively demonstrated in the present study. Earlier stage trainee doctors demonstrated

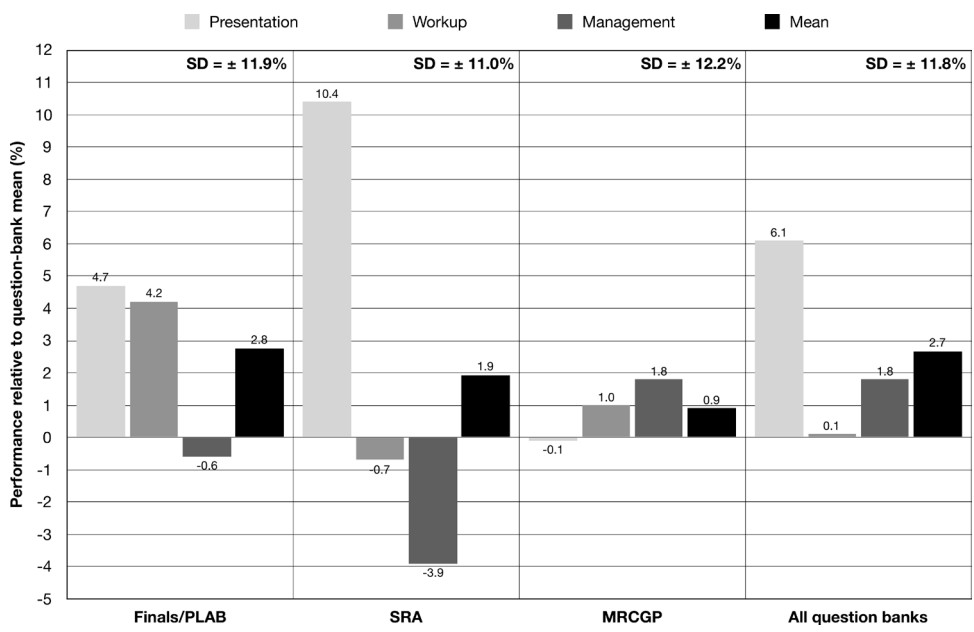

**Figure 1** Trainee knowledge analysis: user performance by degenerative cervical myelopathy (DCM) question theme. DCM questions assessed three themes—presentation, workup and management. Early stage trainees, those sitting finals/PLAB and the SRA, performed worse in questions relating to management versus more senior trainees taking the MRCGP. Performance for MRCGP trainees was more consistent. The average performance for each question theme was within one SD of the mean. MRCGP, Membership of The Royal College of General Practitioners; PLAB, Professional and Linguistic Assessments Board; SRA, Specialty Recruitment Assessment.

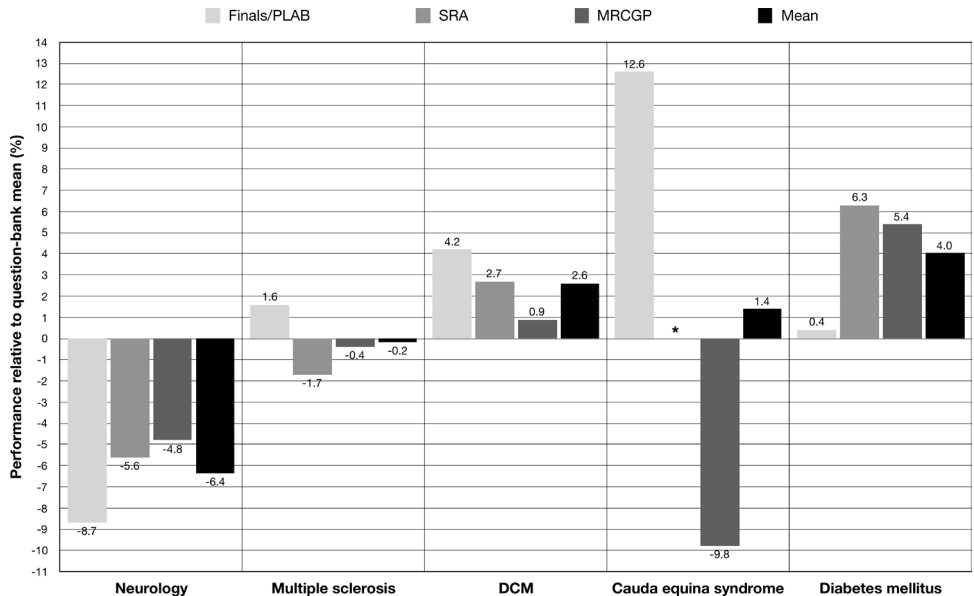

**Figure 2** Trainee knowledge analysis: user performance by disease. The mean user performance for DCM, neurology and cauda equina syndrome decreased with advancing question-bank. This trend was observed in most neurological pathologies. User performance of diabetes mellitus was more variable. *There were no questions on cauda equina syndrome present in the SRA question-bank. DCM, degenerative cervical myelopathy; MRCGP, Membership of The Royal College of General Practitioners; PLAB, Professional and Linguistic Assessments Board; SRA, Specialty Recruitment Assessment.

comparably higher levels of knowledge with regards to the presentation and work up of DCM and a potential gap with regards to management-related questions. These differences disappeared in individuals studying for the GP exit examinations. Nevertheless, the clinical argument for an education gap is strong—our previous analysis of the referral pathway for DCM identified a time to initial referral by the GP as representing 51% of the overall diagnostic delay.

This study demonstrated DCM received relatively less training than other conditions. In the west, the prevalence of DCM is underestimated but at least 60 per 100 000, compared with around 100 per 100 000 for MS.[21 22] Despite this however, DCM was under-represented by eightfold compared with MS in curricula analysis. In late stage curricula (foundation programme, MRCGP), DCM was not referred to at all. A similar trend was observed in textbook analysis, though neither of these analyses were exhaustive. These results were in contrast to our knowledge analysis, which showed consistently above average performance for DCM unlike MS.

One potential way of reconciling the absence of a gap in formal knowledge with the deficiencies in the diagnosis and management of DCM, is that factual knowledge may not translate into recognition of the disease in the clinical context. Causal association between training and clinical practice have not been clearly demonstrated, although prior studies have found good correlation between examination performance and clinical performance in a variety of medical and surgical specialties.[27–29] Most of this data are based on the US Medical Licensing Examination, which appears to correlate well with residency performance.[27 28] Beyond medical school examinations,

in another study, internal medicine clinicians failing their maintenance of certification examinations had a more than double chance of disciplinary action versus those who passed.[29] An alternative explanation is that questionnaire based methods may not be sufficiently sensitive to detect poor clinical decision making in the context of DCM. Future studies should consider employing mock patients for the condition in undergraduate and postgraduate OSCEs, to compare performance to other conditions.

There are several limitations to be considered in the context of these findings. First, only select learning resources were used in this study and questions were inserted into a single online question-bank. Second, question-bank data were not extractable per user, nor per first time answer. While this was not a comprehensive strategy, this is unlikely to have limited our results for at least two reasons: questions were attempted more than 100 000 times and analysis was relative within learning resources, and therefore limitations were applicable to each comparator. The native question-bank data extraction technique also did not provide data on the number of attempts on question themes other than DCM.

The methodology employed in the present study may not have been sufficiently sensitive to detect a knowledge gap for DCM, for several reasons. First, our questions may not have been challenging enough given that they tested core principles all clinicians would be expected to achieve. Although we did not employ pilot testing of the questions in our target population, questions were designed by an experienced author panel including educationalists. Furthermore, our questions were subject to additional scrutiny by the question-bank editors, such

that only those questions deemed appropriate were included in a particular question-bank. Second, there was no first-time answer data available for analysis. This means that 'random effect answering', users selecting an answer to read the explanation, was not accounted for, though this was the case for both DCM and controls. Third, our controls may not have been appropriate and certainly, educational comparisons between specialties is not always the best method of assessing knowledge adequacy.[30] Finally, our methodology assumed a direct correlation between 'on paper' knowledge and clinical care provided to patients—though this was not proven in the present study and could still differ vastly.

## CONCLUSION

In this study, we set out to evaluate whether there is a deficiency in training and knowledge regarding DCM in the GP training pathway. Although DCM was infrequently referenced in learning resources, trainees performed above average for DCM questions in an assessment of knowledge using an online question-bank. The clinical suggestion of an education gap is strong, and these contrasting findings do not conclusively allay this concern. Future studies are required to better understand these observations.

**Acknowledgements** The authors would like to thank Passmedicine for their assistance with this work. The authors acknowledge that research in the senior author's laboratory is supported by a core support grant from the Wellcome Trust (097922/Z/11/Z) and MRC.

**Contributors** MW: coordinated creation of questions, performed statistical analysis, wrote up all manuscript drafts. JW: created first draft of questions for question-bank, aided in manuscript analysis and write-up, helped to check of final draft. JG: aided in creation of first draft of questions for question-bank, aided in manuscript analysis and checking of final draft. BD: checked questions, oversaw data analysis, aided in authoring introduction and collecting key references, helped to devise project idea, checked final manuscript draft. MK: devised study idea, provided draft protocol, checked questions, edited final manuscript draft.

**Funding** Research in the senior author's laboratory is supported by a core support grant from the Wellcome Trust (097922/Z/11/Z) and MRC to the Wellcome Trust-Medical Research Council Cambridge Stem Cell Institute. MK is supported by a NIHR Clinician Scientist Award.

**Disclaimer** This report is independent research arising from a Clinician Scientist Award, CS-2015-15-023, supported by the National Institute for Health Research. The views expressed in this publication are those of the authors and not necessarily those of the NHS, the National Institute for Health Research or the Department of Health and Social Care.

**Competing interests** Research in the senior author's laboratory is supported by a core support grant from the Wellcome Trust and MRC to the Wellcome Trust-Medical Research Council Cambridge Stem Cell Institute. MK is supported by a NIHR Clinician Scientist Award.

**Ethics approval** Not required.

**Provenance and peer review** Not commissioned; externally peer reviewed.

**Data availability statement** All data relevant to the study are included in the article or uploaded as supplementary information.

**ORCID iD**
Mueez Waqar http://orcid.org/0000-0002-7848-6237

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
