## [Reviewer comments · BMJ Open]

ARTICLE DETAILS

TITLE (PROVISIONAL)	A quantitative analysis of medical students' and physicians' knowledge of degenerative cervical myelopathy
AUTHORS	Waqar, Mueez; Wilcock, Jane; Garner, Jayne; Davies, Benjamin; Kotter, Mark

VERSION 1 - REVIEW

REVIEWER	Sarah McCartney Nuffield Orthopaedic Centre, Oxford, UK
REVIEW RETURNED	06-Feb-2019

GENERAL COMMENTS	Thank you for the opportunity to review this interesting paper evaluating a possible knowledge gap amongst GPs in degenerative cervical myelopathy (DCM). The paper adopts an interesting methodology. As I understand it, the authors have reviewed syllabi from both undergraduate and postgraduate training examining the number of references particular conditions enabling a comparison regarding the coverage of DCM in the curriculum. In addition, questions have been inserted into an online question bank used by doctors at various stages of their training. The authors present results that suggest there is paucity of curricula coverage related to DCM across the training pathway but no knowledge gap is evident. This is a novel methodology examining an interesting hypothesis and although the results do not support the stated hypothesis, the authors conclude by arguing that their hypothesis may still be correct but deficiencies in the methodology limit the robustness of the findings. With this in mind, I would suggest the discussion needs some revision prior to publication which should perhaps focus more on the methodology and how it might be evolved to understand this topic further. One of the key questions is how the questions were standard set prior to inclusion in the question bank and how the level of difficulty of these new DCM questions compared to the question bank average and the existing questions within the bank. One might argue that performance on the DCM questions was 'above average' because the questions were in effect easier. There is little discussion of this and this may have had a major influence on the findings. It would be helpful if the authors could clarify this further and if no standard setting was undertaken, explain the rationale for this.
--

	The authors do acknowledge that repeat answers cannot be accounted for, and again this may have had an influence on the results, when compared to the overall question bank mean. How does the number of attempts made at DCM questions compare to the number of attempts at questions in one of the comparator groups, say Diabetes Mellitus. This may give some indication of whether users are repeating difficult questions which is affecting the reported success rate. Of most interest is 1st time success rate, which the study is not able to present. Attempts beyond that may represent learning from the questions. As a minor point, it is probably also worth mentioning any effect 'random answering' may have on the results – users blindly answering questions which they do not have the knowledge to answer in order to read the explanation and discussion. There are areas of the discussion that could also be a little clearer. The first paragraph states 'user performance in DCM questions remained consistently above question-bank averages' but further on in the same paragraph states 'user performance was either consistently below average or decreased sequentially for DCM'. It would be helpful if the authors could clarify what they are concluding in this section. Finally, I am not clear how this study adds to literature surrounding 'neurophobia' (p 15, third paragraph). In Figure 2 the authors present performance in DCM and the identified comparator topics, however, this figure also presents a 'Neurology' comparison which is not described elsewhere in the text, and it is unclear what this is presenting. The performance across the neurological conditions identified (DCM, MS and Cauda Equina) is variable but certainly doesn't demonstrate a 'neurophobia' and I would suggest in places the results disapprove the theory of a neurophobia. As a side note, I am uncomfortable about the propagation of the term 'neurophobia'. I agree it is important to recognise and acknowledge gaps in knowledge and lack of confidence in disease areas but I think care has to be taken not to use terms which might be considered denigrating. A 'poor' knowledge of a particular area does not necessarily represent a 'phobia' on the part of the clinician and as indeed your paper hypothesis may simply reflect inadequate curriculum exposure. Recognising a lack of confidence, is however important. Minor points: In the second paragraph of the introduction on page 6 there is a figure quoted for 'time to initial referral by GP (6.4 +/- 7.7)'. What are the units for this? The wording in the results of the text book analysis (p13) is imprecise. You report that the word count attributed to diabetes mellitus 'seemed' to increase.
--	---

REVIEWER	Jörg Krebs Clinical Trial Unit Swiss Paraplegic Centre Switzerland
REVIEW RETURNED	08-Mar-2019

GENERAL COMMENTS	Review of "A quantitative analysis of medical students' and physicians' knowledge of degenerative cervical myelopathy" – bmjopen-2018-028455
---

	The authors have quantitatively assessed the training and knowledge of medical students and GP trainees regarding presentation, workup and management of DCM compared to other neurological conditions (MS, cauda equine) and diabetes. Even though DCM was referenced infrequently in training resources, knowledge regarding DCM was above average with lower performance regarding the management of DCM. The authors concluded that the study results did not dispel the concern regarding a DCM education gap. This is a relevant and well performed study. Abstract Results  - line 44/45: “joint (?) lowest modal rank”: please rephrase - lines 48-51: “performance decreased with advancing question-bank”: is this statement relevant? Is not more relevant to report that performance was better for clinical presentation than for management? - lines 48/49 and 54/55: please report values for user performance - line 58/59: units of reported values are missing Strengths and Limitations  - identify strengths and limitations - line 28/29: avoid colloquial expressions (i.e. handful) Introduction  - page 6, line 46: please add unit to time value - page 6, line 59: refrain from using “significant”, if it is not in the context of a statistical test – use e.g. “relevant” instead - page 7, lines 12-17: please consider rewriting the sentence, statement in the sentence is blurry Methods  - Tables: legends are missing - Table 2: why is the search term “cervical myelopathy” listed twice? Results  - page 14, line 40 to page 15, line 7: report average performance values and standard deviations
--	---

VERSION 1 – AUTHOR RESPONSE

Reviewer 1 comments

1. I would suggest the discussion needs some revision prior to publication which should perhaps focus more on the methodology and how it might be evolved to understand this topic further.

We have now added a section on future studies and how they could better test our study hypothesis in the discussion as follows:

“...An alternative explanation is that questionnaire based methods may not be sufficiently sensitive to detect poor clinical decision making in the context of DCM. Future studies should consider employing mock patients for the condition in undergraduate and postgraduate OSCEs, to compare performance to other conditions.”

2. One might argue that performance on the DCM questions was 'above average' because the questions were in effect easier. There is little discussion of this and this may have had a major influence on the findings. It would be helpful if the authors could clarify this further and if no standard setting was undertaken, explain the rationale for this.

We have expanded on this in the limitations section of the paper:

"Although we did not employ pilot testing of the questions in our target population, questions were designed by an experienced author panel including educationalists. Furthermore, our questions were subject to additional scrutiny by the question-bank editors, such that only those questions deemed appropriate were included in a particular question-bank."

3. How does the number of attempts made at DCM questions compare to the number of attempts at questions in one of the comparator groups, say Diabetes Mellitus.

Unfortunately, despite our enquiries, the question bank platform were unable to provide this information. We have acknowledged this in our limitations as follows:

"The native question bank data extraction technique also did not provide data on the number of attempts on question themes other than DCM."

4. As a minor point, it is probably also worth mentioning any effect 'random answering' may have on the results – users blindly answering questions which they do not have the knowledge to answer in order to read the explanation and discussion.

We have added this to the limitations section:

"Secondly, there was no first-time answer data available for analysis. This means that 'random effect answering', users selecting an answer to read the explanation, was not accounted for, though this was the case for both DCM and controls."

5. There are areas of the discussion that could also be a little clearer. The first paragraph states 'user performance in DCM questions remained consistently above question-bank averages' but further on in the same paragraph states 'user performance was either consistently below average or decreased sequentially for DCM'. It would be helpful if the authors could clarify what they are concluding in this section.

We have reworded the final sentence to avoid this conflict as follows.

“There was a sequential decrease in user performance across the themes of DCM presentation, workup and management for early years’ trainees, whereas for senior trainees, performance did not vary by theme of question.”

6. Finally, I am not clear how this study adds to literature surrounding 'neurophobia' (p 15, third paragraph). In Figure 2 the authors present performance in DCM and the identified comparator topics, however, this figure also presents a 'Neurology' comparison which is not described elsewhere in the text, and it is unclear what this is presenting. The performance across the neurological conditions identified (DCM, MS and Cauda Equina) is variable but certainly doesn't demonstrate a 'neurophobia' and I would suggest in places the results disapprove the theory of a neurophobia. As a side note, I am uncomfortable about the propagation of the term 'neurophobia'. I agree it is important to recognise and acknowledge gaps in knowledge and lack of confidence in disease areas but I think care has to be taken not to use terms which might be considered denigrating. A 'poor' knowledge of a particular area does not necessarily represent a 'phobia' on the part of the clinician and as indeed your paper hypothesis may simply reflect inadequate curriculum exposure. Recognising a lack of confidence, is however important.

We have defined the neurology questions within the methods section:

“The question-bank also provided data on neurology as a theme, encompassing all questions relating to the central and peripheral nervous system.”

We have now reworded the paragraph in our discussion as follows:

“We observed a below average performance in questions grouped under the neurology theme, which included all questions relating to the central and peripheral nervous system. Reduced knowledge pertaining to neurosciences has previously been linked to a term called neurophobia, though this is by no means a universally accepted concept.^{8, 9, 23} In one questionnaire study for example, GPs rated neurology as the most difficulty medical specialty and the one for which they had the least confidence compared to cardiology, endocrinology, gastroenterology, geriatrics, respiratory medicine and rheumatology.⁹ However, the question-bank data did not allow distinction between basic science and clinical questions, for whom the performance may be different, as evidenced by the above average performance in our clinically orientated DCM questions.”

7. In the second paragraph of the introduction on page 6 there is a figure quoted for 'time to initial referral by GP (6.4 +/- 7.7)'. What are the units for this?

This has been corrected as follows.

“Our analysis of this pathway has identified time to initial referral by GP (6.4±7.7 months) as representing 51% of diagnostic delay...”

8. The wording in the results of the text book analysis (p13) is imprecise. You report that the word count attributed to diabetes mellitus ‘seemed’ to increase.

The word ‘seemed’ has been removed.

Reviewer 2 comments

1. Abstract: line 44/45: “joint (?) lowest modal rank”: please rephrase

We have removed the word joint and also added to the abstract method about how a comparison was performed using modal ranks.

“Assessment of training: quantitative comparison of references to DCM in curricula (undergraduate/postgraduate) and commonly used textbooks (Oxford Handbook Series), to other conditions using modal ranks.”

2. Abstract: lines 48-51: “performance decreased with advancing question-bank”: is this statement relevant? Is not more relevant to report that performance was better for clinical presentation than for management?

We have removed this sentence. The latter is already included in the last line of the results section with relevant data as follows.

“Performance for DCM questions in themes of presentation (+6.1%) , workup (+0.1%) and management (+1.8%) were all greater than the question-bank mean and within one standard deviation.”

3. Abstract: lines 48/49 and 54/55: please report values for user performance, and Abstract: line 58/59: units of reported values are missing

These have been added:

“For students and junior trainees, there was a serial decrease in performance from presentation and workup (-0.7% to +10.4% relative to question-bank mean) and management (-0.6% to -3.9% relative to question-bank mean).”

4. Strengths and Limitations: identify strengths and limitations

We have now specified strengths and limitations:

- Strength: Search terms relating to DCM were queried from three UK specific medical school curricula and relevant postgraduate curricula
- Strength: a large number of responses were obtained by placing questions in an online question-bank, relating to DCM
- Limitation: A limited number of learning resources were searched to assess references to DCM

5. Strengths and Limitations: line 28/29: avoid colloquial expressions (i.e. handful)

This is corrected

“Limitation: A limited number of learning resources were searched to assess references to DCM”

6. Introduction: page 6, line 46: please add unit to time value

This has been corrected as follows.

“Our analysis of this pathway has identified time to initial referral by GP (6.4 ± 7.7 months) as representing 51% of diagnostic delay...”

7. Introduction: page 6, line 59: refrain from using “significant”, if it is not in the context of a statistical test – use e.g. “relevant” instead

This has been changed:.

“This period of the diagnostic pathway is difficult to examine in detail, and whilst delayed patient presentation is likely to contribute, delayed detection measured by multiple consultations and patient perspective, is certainly a relevant component.”

8. Page 7, lines 12-17: please consider rewriting the sentence, statement in the sentence is blurry

This has been changed:

Comparator diseases were also selected to compare findings to:

1. Direct comparator to myelopathy: a disease that is a differential diagnosis for DCM with equivalent or greater incidence.
2. Degenerative spine comparator: an alternative degenerative spine disease that is widely taught.
3. Generic non-neuroscience comparator: a common disease that all clinicians would have some knowledge about and interaction with.

9. Tables: legends are missing

The legend for table 1 has been added. Other legends are present and have been reviewed:

Table 1. Summary of gap analysis methods. This table shows the methods used in this study. Curricula and textbooks were screened by training stage to assess references to key search terms. An online question bank was used in knowledge assessment.

10. Table 2: why is the search term “cervical myelopathy” listed twice?

This was an error and has been removed

11. Results: page 14, line 40 to page 15, line 7: report average performance values and standard deviations

As our results are based on reporting these values relative to the question-bank mean, these adjusted values have now been added, as also shown in the figures. The original non-adjusted values are provided in the supplementary table. Providing the non-adjusted values would add a degree of confusion to the results.

“There were differences in user performance in the three DCM question themes – presentation, workup and management, though average scores were all within 1 standard deviation of the mean (Fig. 1). Performance sequentially decreased across these themes for the finals/PLAB group (presentation +4.7%, workup +4.2%, management -0.6%; relative to question-bank mean), SRA group (presentation +10.4%, workup -0.7%, management -3.9%; relative to question-bank mean) and when all groups were considered together (presentation +6.1%, workup +0.1%, management +1.8%; relative to question-bank mean). In the MRCGP group however, performance clustered around the mean for the question bank with little variability (presentation -0.1%, workup +1.0%, management +1.8%; relative to question-bank mean).

User performance by disease showed large variation between question-banks (Fig. 2). The mean user performance decreased sequentially for DCM with advancing question-bank (finals/PLAB +4.2%, SRA +2.7%, MRCGP +0.9%; relative to question-bank mean). However, the mean user performance for DCM was always greater than the question-bank mean (+2.6% relative to question-bank mean).”

VERSION 2 – REVIEW

REVIEWER	Sarah McCartney Specialty Trainee in Trauma and Orthopaedics, Nuffield Orthopaedic Centre, Oxford, UK
REVIEW RETURNED	06-May-2019

GENERAL COMMENTS	Many thanks for the opportunity to review the revised version of this manuscript. It addresses an important issue and now describes and discusses the methodology much more clearly.
--

REVIEWER	Jörg Krebs Clinical Trial Unit Swiss Paraplegic Centre Nottwil, Switzerland
REVIEW RETURNED	01-May-2019

GENERAL COMMENTS	The authors have adequately addressed all issues raised by the reviewers.
---